# Effect of Ultrasonic Excitation on Discharge Performance of a Button Zinc–Air Battery

**DOI:** 10.3390/mi12070792

**Published:** 2021-07-02

**Authors:** Zhao Luo, Qiang Tang, Junhui Hu

**Affiliations:** 1State Key Lab of Mechanics and Control of Mechanical Structures, Nanjing University of Aeronautics and Astronautics, Nanjing 210016, China; luozhao@nuaa.edu.cn; 2Jiangsu Provincial Engineering Research Center for Biomedical Materials and Advanced Medical Devices, Faculty of Mechanical and Material Engineering, Huaiyin Institute of Technology, Huaian 223003, China; tangqiang102@126.com

**Keywords:** zinc–air battery, ultrasonic excitation, acoustofluidic field, performance improvement

## Abstract

In this paper, a method to increase the output power of a button zinc–air battery by applying acoustofluidics induced by ultrasonic excitation to the battery is proposed and demonstrated. In the structural design of the device, a flat piezoelectric ring was bonded onto the top of the outer surface of the cathode shell to excite an ultrasonic field in the battery. The maximum output power of the zinc–air battery increased by 46.8% when the vibration velocity and working frequency were 52.8 mm/s (the corresponding vibration amplitude was 277 nm) and 161.2 kHz and the rating capacity increased by about 20% with the assistance of the acoustofluidic field induced by ultrasonic excitation. Further analyses indicated that the discharge performance improvement can be attributed to the acoustic microstreaming vortices and the decrease of the viscosity coefficient in the electrolyte solution, which were both caused by ultrasonic excitation of the piezoelectric ring.

## 1. Introduction

Owing to the merits such as high energy density (1086 W·h/kg in theory), eco-friendliness, low cost, and high safety [1,2,3], zinc–air batteries have garnered extensive interest in the energy engineering field over recent years. Therefore, a series of studies on high-performance zinc–air batteries have been carried out [4,5], with most focusing on the development of high-performance electrocatalysts [6], such as the adoption of precious metals and alloys [7,8,9], hetero-atoms doped casrbonaceous materials [10,11,12,13], perovskite materials [14,15,16], and transition metal carbides and oxides [17,18]. Efforts also have been made in the optimization of zinc species allocation in the zinc electrode, which include utilization of high-surface-area porous structures [19,20], 3D conductive host materials [21,22,23], etc. In addition to the above methods, the electrolyte [24,25], separator [26,27,28], and other components [29,30] of zinc–air batteries have also been studied to improve their performance.

Up to the present, with the development of zinc–air battery performance, the primary zinc–air batteries have been commercially implemented for some applications such as hearing-aids and outdoor lighting [6,31,32]. However, despite the primary commercialization and promising applications, the development of zinc–air batteries has been impeded by some limitations associated with the metal and air electrodes, and the current capability of existing zinc–air batteries is far from satisfactory [1,33,34]. Urgent issues which need to be solved in the development of high-performance zinc–air batteries include slow ORR (oxygen reduction reaction) and OER (oxygen evolution reaction) kinetics [35], non-uniform zinc dissolution and deposition [36], the extended cycle life required [37], and electrolyte reaction with CO_2_ to form carbonates [38], etc.

In this paper, a method to enhance the output power and increase the rating capacity of a commercialized button zinc–air battery is proposed and demonstrated. In the structural design of the device, a piezoelectric ring was used as a vibration source, which was bonded to the top of the battery to excite an ultrasonic field inside the battery. The experimental results show that the output power of the battery increased from 22.2 mW to 32.6 mW, and the rating capacity increased from 330.56 mA·h to 396.89 mA·h when the vibration velocity is 52.8 mm/s (the corresponding vibration amplitude was 277 nm) at 161.2 kHz. The vibration velocity was the averaged peak–peak value of the out-of-plane vibration velocity on the upper surface of piezoelectric ring. In addition, the AC (alternating current) impedance of the battery was measured and computed to analyze the working principle. The analyses showed that the ultrasonic effects in electrolyte solution, such as acoustic microstreaming vortices and viscosity decrease, contribute to the discharge performance improvement by enhancing the uniformity of OH^−^ distribution and decreasing the resistance of mass transfer [37,39].

## 2. Results, Analyses, and Discussion

The front and back views of the button zinc–air battery are shown in Figure 1a,b. A piezoelectric ring of 6 mm (inner diameter) × 12 mm (outer diameter) × 8 mm (thickness) was bonded on the top of the outer surface of air electrode of the battery (A675/PR44, Fujian Nanping Nanfu Battery Co., Ltd., Nanping, China). There were six air inlets in the cathode shell. One is at the center of the top surface, and the rest were distributed on a circumference with a diameter of 5 mm. The inner diameter of the piezoelectric ring was chosen to be 6 mm, to ensure the air inlets were not covered by the piezoelectric ring. During the operation procedure, the piezoelectric ring acted as an oscillation source to mechanically excite the battery by applying an AC voltage from a power amplifier (HFVP-83A, Nanjing Foneng Science and Technology Industrial Co., Ltd., Nanjing, China), which received a sinusoidal signal generated by a function generator (ARG 3022B, Tektronix, Tektronix China Ltd., Shanghai, China). The above-mentioned function generator and power amplifier were used only for the convenience of our experimental process, and the electrical driving system for the piezoelectric ring can be miniaturized in practical applications. The driving voltage and current of the piezoelectric ring and the phase difference between them were measured and monitored by an oscilloscope (DPO2014, Tektronix, Tektronix China Ltd., Shanghai, China). The vibration distribution of the piezoelectric ring was measured by a 3D laser Doppler vibrometer (PSV-500, Polytec Ltd., Berlin, Germany). Figure 1c shows the detailed structure of the button zinc–air battery with a piezoelectric ring. A Teflon^®^ air diffusion layer (polytetrafluoroethylene), Teflon^®^ hydrophobic layer (Polytetrafluoroethylene), wire mesh framework with catalytic materials layer (nickel-plated wire mesh), and diaphragm (polyamide) were stacked beneath the end surface of the cathode shell. The inner cavity of the battery, which was formed by the cathode and anode shells, was full of electrolyte mixed with zinc powders. Insulating material was filled in between the anode and cathode shells to prevent a short circuit.

The electrode reactions are expressed as follows.

Cathode: O_2_ + 2H_2_O + 4e^−^ → 4OH^−^

Anode: Zn + 4OH^−^ → Zn(OH)_4_^2−^ + 2e^−^

Zn(OH)_4_^2−^ → ZnO + H_2_O + 2OH^−^

Overall reaction: 2Zn + O_2_ → 2ZnO

Parasitic reaction: Zn + 2H_2_O → Zn(OH)_2_ + H_2_

Figure 2 shows the measured characteristics of the button zinc–air battery ultrasonically excited under different vibration velocities at 161.2 kHz. Figure 2a shows the relationship between the output voltage and current of the battery, and Figure 2b shows the relationship between the output power and current of the battery. An adjustable resistor (Adjustable Resistance Box ZX21a, Tianshui Great Wall Electrical Instrument Co., Ltd., Tianshui, China) is used as the load of the battery, and the values of resistance in the experiment are from 5 Ω to 1000 Ω, including 5, 10, 15, 20, 25, 30, 35, 40, 50, 60, 70, 100, 200, 500, and 1000 Ω. The output voltage was measured by a digital multimeter (FLUKE 8845A) under different excitation conditions. The abbreviation *v*_p-p_ in the figure is the averaged peak–peak value of the out-of-plane vibration velocity on the upper surface of the piezoelectric ring, which was measured by the 3D laser Doppler vibrometer. For comparison, the results when there is no ultrasonic excitation (*v*_p-p_ = 0 mm/s) are also given in the figure. It can be observed that for a given discharge current, the ultrasonic excitation could increase the output voltage and power. In addition, the effect became more obvious with the increase of the discharge current. When the vibration velocity *v*_p-p_ was 52.8 mm/s, the maximum output power of the battery was 32.6 mW, which was 48% higher than that without ultrasonic excitation (22.2 mW).

In order to elucidate the mechanism of discharge-performance-improvement of the button zinc–air button battery excited by ultrasonic vibration, the AC impedance of the battery is measured by an electrochemical workstation (CHI760E, Shanghai Chen Hua Co., Ltd., Shanghai, China), and the results are shown in Figure 3. Figure 3a shows the Nyquist plots under different vibration velocities, indicating that both the resistance and capacitive reactance decrease with the increase of ultrasonic vibration velocity. The inset in Figure 3a shows that the ohmic resistance *R*_s_, which is the intercept with the *X*-axis (at high frequency) and consists of the electrolyte solution resistance and contact resistance, has a slight decrease with the increase of ultrasonic vibration velocity. Figure 3b shows the Bode plots under different vibration velocities. It is known that the impedance in the low-frequency range in a Bode plot is mainly caused by the concentration polarization and that in the middle frequency range is mainly caused by the electric double layer effect [39]. Therefore, Figure 3b indicates that ultrasonic excitation affects both the concentration polarization and electric double layer effect, and the effect on the former is larger than that on the latter.

It is known that the diffusion layer thickness *δ* in the forced convection is related to the diffusivity *D*, electrolyte kinetic viscosity *μ**_f_*, and flow velocity *u*_0_ of electrolyte [40]. The diffusion layer thickness *δ* decreases as the electrolyte flow velocity *u*_0_ increases, and the electrolyte kinetic viscosity *μ**_f_* decreases [41]. Considering the propagation of the ultrasonic field into the inner cavity of the battery, the weakening of both concentration polarization and electric double layer effect can be attributed to the acoustic microstreaming vortices and viscosity decrease of the electrolyte solution induced by the acoustofluidic field [40,42]. The acoustic microstreaming vortices can flush the ions in the electrolyte solution away from the electric double layer and weakens the electric double layer effect. The slight decrease of the ohmic resistance due to ultrasonic excitation is attributed to the viscosity decrease induced by ultrasound.

To confirm the existence of the ultrasonic field and acoustic streaming vortices in the battery, the vibration of the piezoelectric ring was measured and the acoustofluidic field in the inner cavity of the battery was computed by the commercial FEM software COMSOL Multiphysics (version 5.4, COMSOL AB, Stockholm, Sweden). Unless otherwise specified, all material parameters used in the simulation process in the simulation are listed in Table 1. Vibration distributions on the top surface of the piezoelectric ring were measured at 161 kHz and 18 V_p-p_ by the 3D laser Doppler vibrometer, and the measured in-plane and out-of-plane vibration distributions are shown in Figure 4a,b. It is seen that the piezoelectric ring mainly vibrated in the radial direction at the working frequency, which can excite the anode shell to vibrate flexurally. The vibration distribution of the battery and ultrasonic field in the electrolyte can be clarified in the following simulation results.

Figure 4c shows a 3D meshed model of the button zinc–air battery, and Figure 4d shows the computed vibration displacement of the whole shell, in which the color represents the total vibration displacement magnitude. It is seen that the shell of the battery vibrated flexurally with the ultrasonic excitation of the piezoelectric ring. Figure 4e shows the sound pressure distribution in the central plane of the inner cavity of the battery. During the negative half periods of sound pressure, the distance among the molecules in the electrolyte increased, causing that the cohesive forces among the molecules to become small. During the positive half periods of sound pressure, the molecules were compressed by the positive sound pressure. However, the cohesive forces cannot increase too much because the molecules repelled each other when they were too close. Thus during the entire vibration period, the averaged cohesive forces among the liquid molecules become weaker under sonication [42]. As a result, the equivalent viscosity of the electrolyte decreases because of the sound pressure inside the battery, which contributes to the weakening of the concentration polarization and double layer effect. Figure 4f shows the computed acoustic streaming field in the central plane of the inner cavity. The mixing function of the acoustic streaming vortices improves the uniformity of the ion distribution and reduces the concentration polarization and double layer effect [37,40]. Therefore, the internal resistance of the button zinc–air battery reduces, and the output power increases.

More characteristics of the ultrasonically excited button zinc–air battery have been investigated. The influence of the acoustic streaming field on the battery under constant current discharge can be measured by the battery test system BT2018A (Hubei Lanbo New Energy Equipment Co., Ltd., Wuhan, China), and the results are shown in Figure 5a. In the experiments, the vibration velocity was 52.8 mm/s, the working frequency was 161.2 kHz, and the discharge current was 20.3 mA. The capacity *C* was calculated by the following equation
(1)C=I·t
where *I* and *t* are the discharge current and time, respectively. Figure 5a shows that the working time, discharge voltage, and capacity of the battery with the ultrasonic assistance are 19 h, 0.85–1.15 V, and 396.89 mA·h, respectively, and those without the ultrasonic assistance were 16.3 h, 0.75–1.0 V, and 330.56 mA·h, respectively. Thus, with the assistance of acoustofluidic field, the discharge time was extended for about 3 h, and the capacity was increased by 20%.

Figure 5b shows the measured maximum output power of the battery versus vibration velocity at the upper surface of the piezoelectric ring (*r* = 9 mm). It is seen that the maximum output power increase with the increase of ultrasonic vibration velocity. When the vibration velocity is 52.8 mm/s, the maximum output power is 32.6 mW, which is 48% higher than the maximum output power of the battery without ultrasonic assistance (22.2 mW).

Figure 5c shows the measured open-circuit voltage of the button zinc–air battery under different vibration velocities at 161.2 kHz. It can be seen that the open-circuit voltage remains unchanged at around 1.41 V, which means that the ultrasonic vibration has no influence on the open-circuit voltage of zinc–air battery. In order to analyze and explain the above-mentioned influence in Figure 5c, a zinc–air galvanic cell shown in Figure 6a is designed to investigate the ultrasonic effect on the single-electrode potential. The zinc, platinum, and Hg/HgO electrodes are set to be the working, auxiliary, and reference electrodes, respectively, and 30 wt% KOH is set to be the electrolyte. A copper plate is bonded to the radiation surface of a Langevin transducer to transmit vibration into an electrolyte to generate the acoustic streaming vortices. Figure 6b shows the measured potentials of the zinc and platinum electrodes under different vibration velocities of the Langevin transducer’s radiation surface. It was seen that the increase of the vibration velocity has little influence on the anode and cathode potentials, which indicates that the ultrasonic vibration did not affect the single-electrode potential. This is because in the open circuit condition, the redox reaction of the battery did not occur, and the ultrasonic vibration had little influence on the temperature and ion concentration on the electrodes [40]. This procedure was applied when a higher output power was needed. Whether this is a permanent or disposable procedure depends on whether the battery can be charged. In our work, the zinc–air battery is disposable. Once the ultrasonic vibration is switched on, the procedure lasts until the battery runs out of power. The entire procedure can last for nearly 20 h, as shown in Figure 5a.

## 3. Conclusions

A strategy to enhance the output power of a zinc–air battery by ultrasonic vibration is proposed and demonstrated. A piezoelectric ring bonded onto a commercialized button zinc–air battery is used to the produce ultrasonic field inside the battery. There is a 48% increase in the output power of the battery when the vibration velocity is 52.8 mm/s at 161.2 kHz, and the ultrasonic vibration can increase the rating capacity by about 20%. Based on the measured AC impedance and computed acoustofluidic field, the discharge performance improvement is attributed to the acoustic microstreaming vortices and viscosity decrease of the electrolyte solution, caused by ultrasonic excitation. This principle can also be applied to a battery pack to enhance the discharge performance and rating capacity by using a suitable ultrasonic vibration system.

## Figures and Tables

**Figure 1 micromachines-12-00792-f001:**
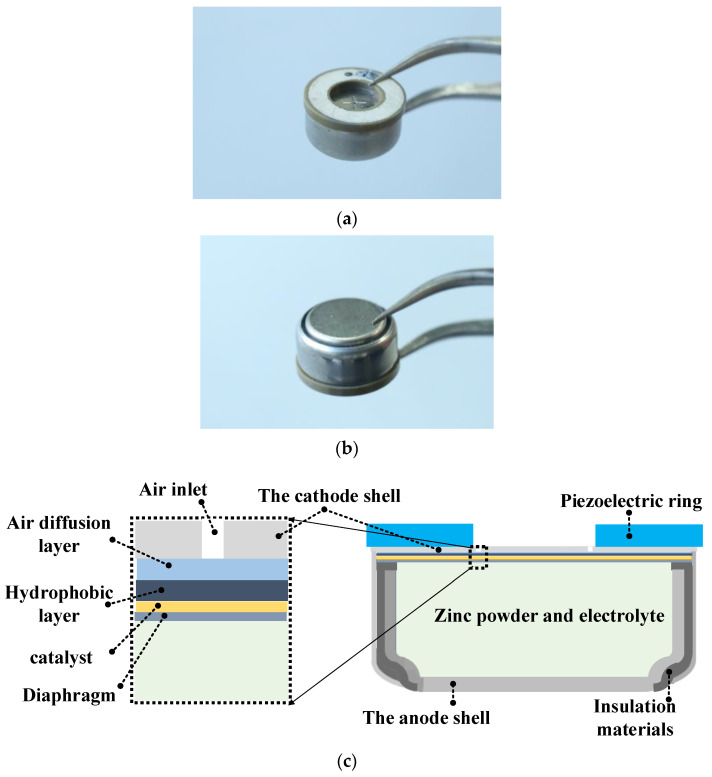
Photos and schematic of the button zinc–air battery with a piezoelectric ring. (**a**) Front view. (**b**) Back view. (**c**) Structural schematic.

**Figure 2 micromachines-12-00792-f002:**
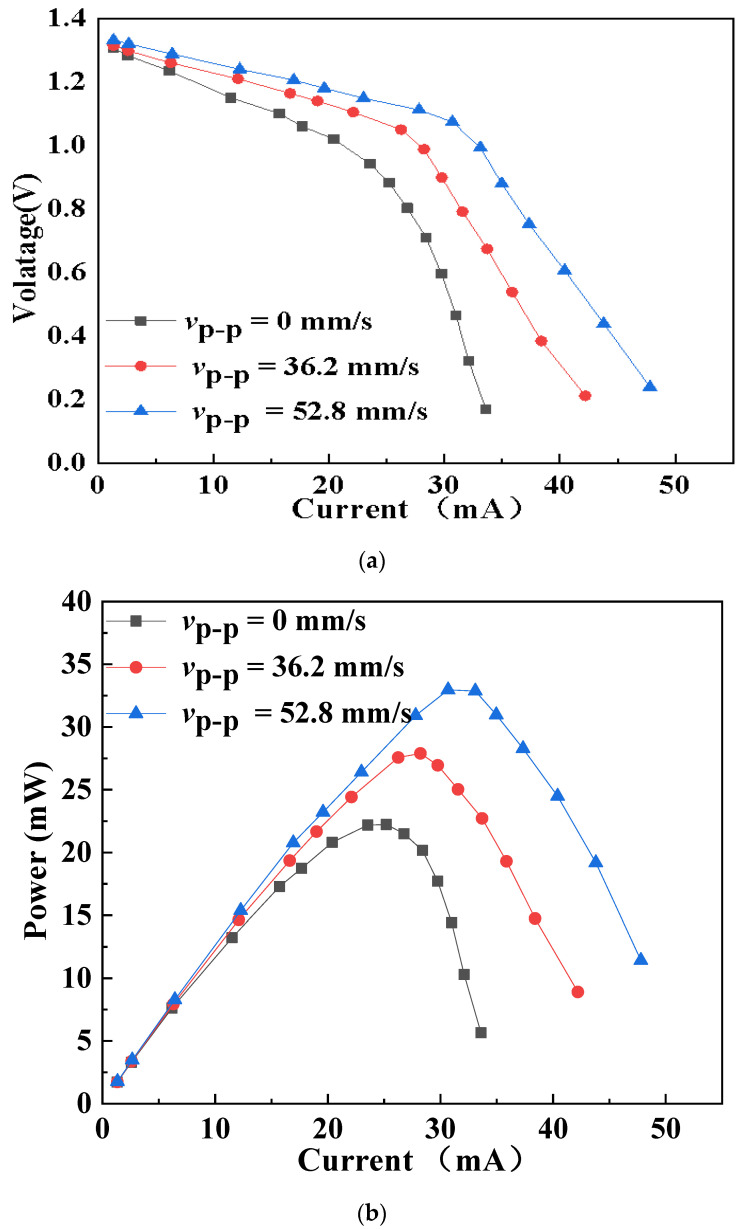
Characteristics of the button zinc–air battery ultrasonically excited under different vibration velocities at 161.2 kHz. (**a**) Voltage vs. current. (**b**) Power vs. current.

**Figure 3 micromachines-12-00792-f003:**
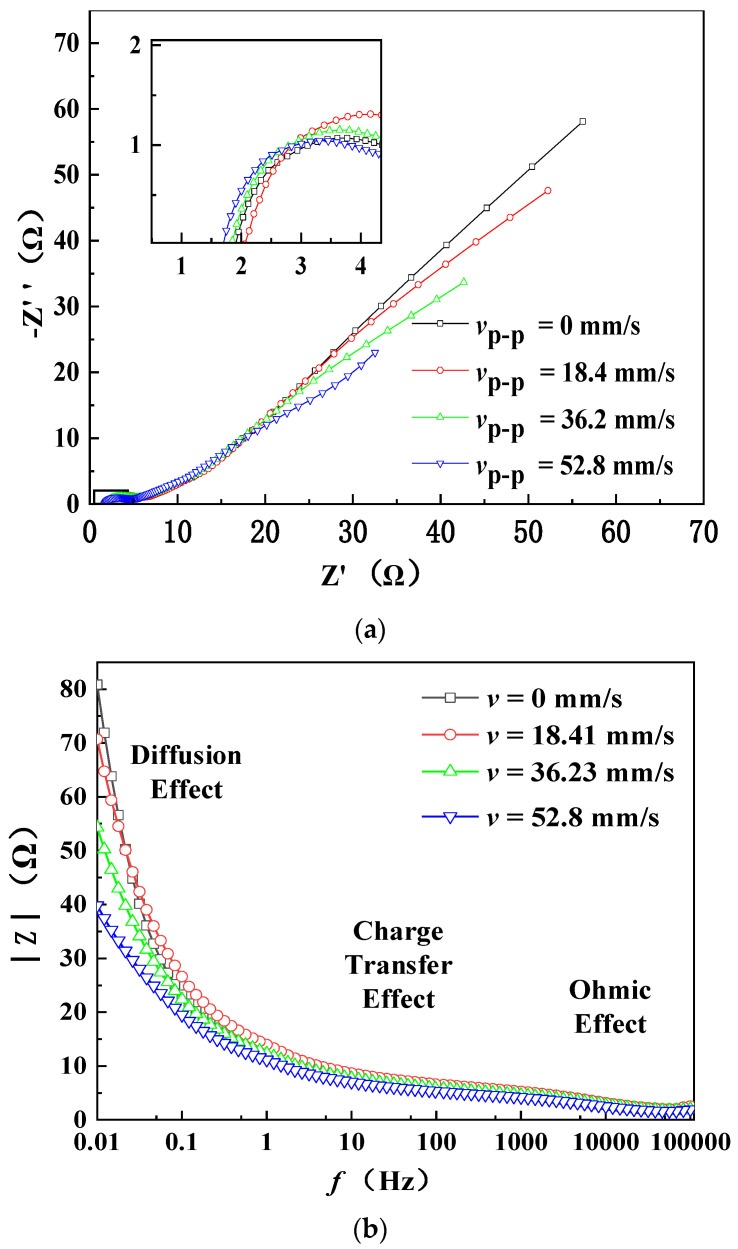
Measured AC impedance of the ultrasonically excited zinc–air battery under different vibration velocities at 161.2 kHz. (**a**) Nyquist plots; (**b**) Bode plots.

**Figure 4 micromachines-12-00792-f004:**
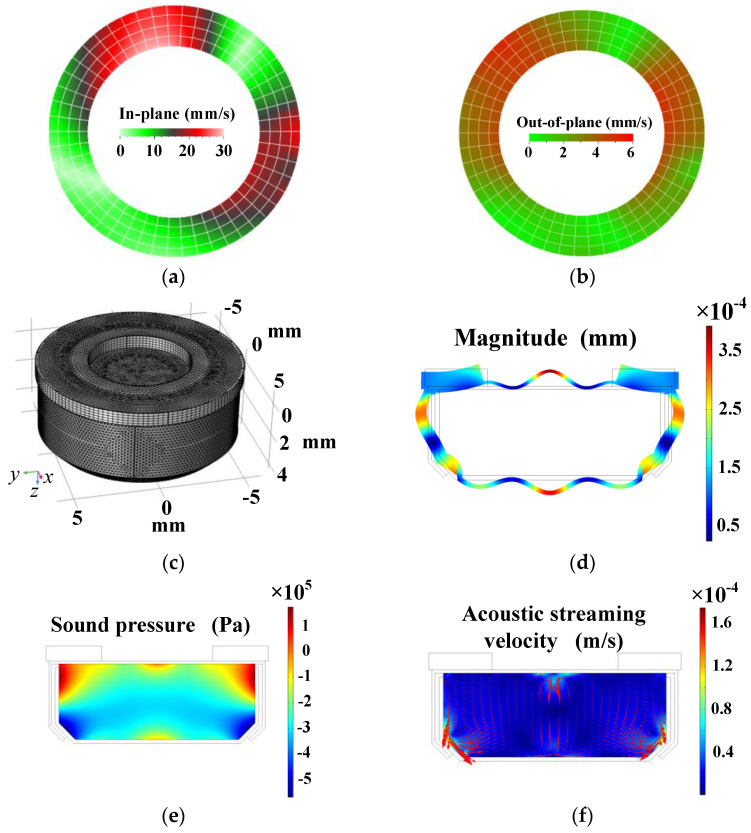
Measured vibration distribution at 161 kHz and 18 *V**_p-p_* and computed acoustofluidic field in the battery. (**a**) The in-plane vibration of the piezoelectric ring’s top surface. (**b**) The out-of-plane vibration of the piezoelectric ring’s top surface. (**c**) A 3D grid meshed model of the battery. (**d**) Vibration displacement pattern of the battery. (**e**) Sound pressure distribution in the central plane of the inner cavity. (**f**) Velocity distribution of acoustic streaming in the central plane of the inner cavity.

**Figure 5 micromachines-12-00792-f005:**
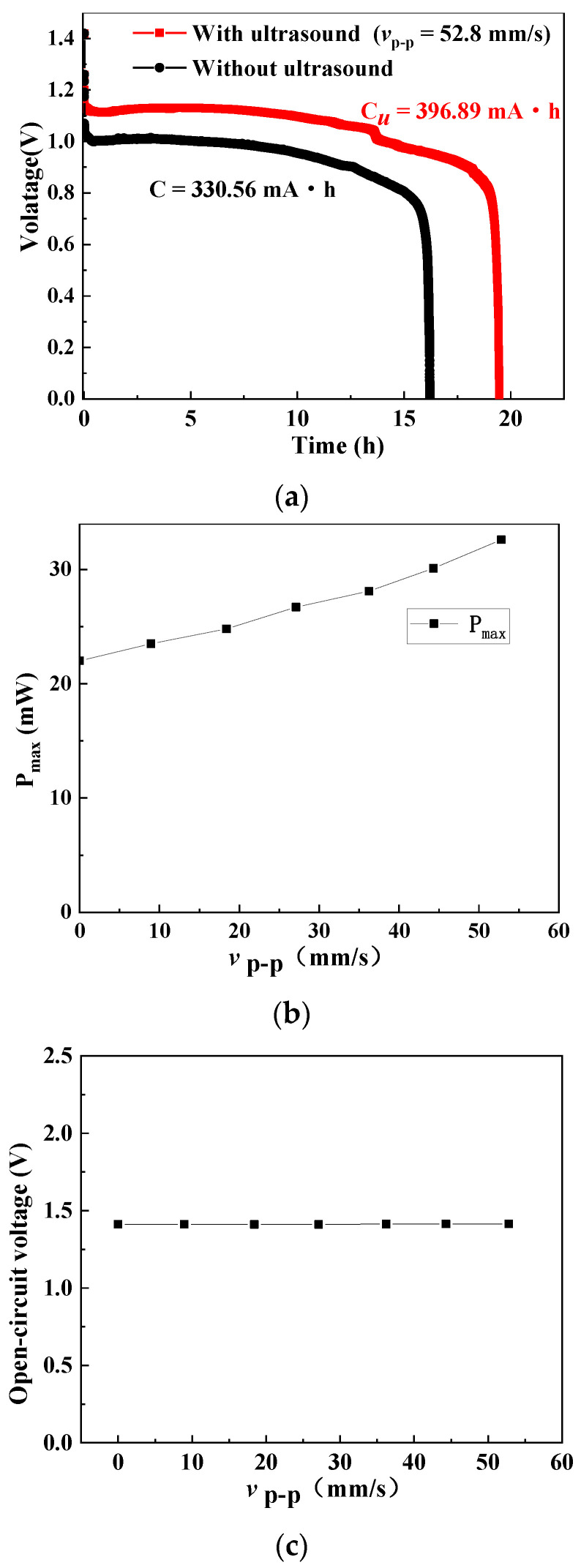
Measured characteristics of the ultrasonically excited button zinc–air battery. (**a**) Discharge profile under a constant discharging current (20.3 mA) with and without sonication. (**b**) Maximum power vs. ultrasonic vibration velocity. (**c**) Open-circuit voltage vs. ultrasonic vibration velocity.

**Figure 6 micromachines-12-00792-f006:**
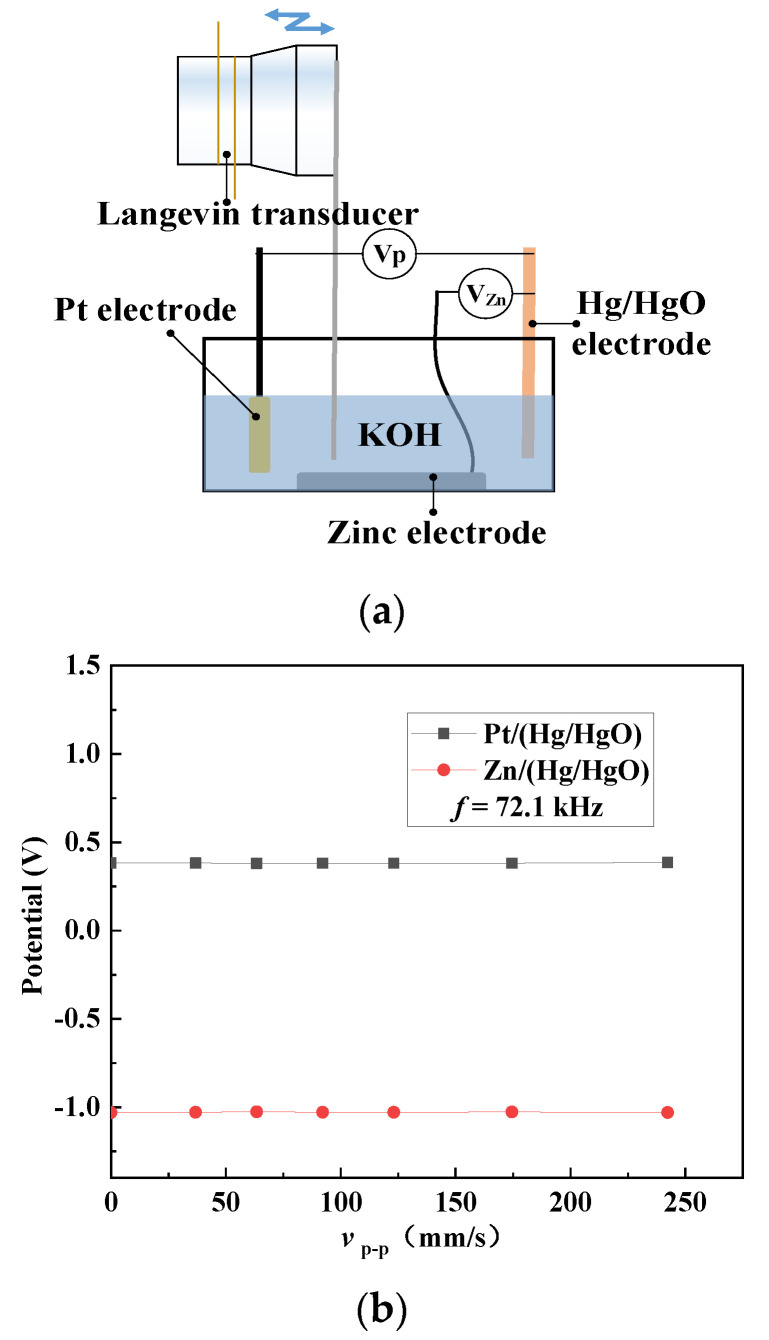
Independency of single-electrode potentials of the zinc–air battery on the electrolyte’s ultrasonic vibration. (**a**) Schematic diagram of testing device; (**b**) Measured Pt/(Hg/HgO) and zinc/(Hg/HgO) potentials.

**Table 1 micromachines-12-00792-t001:** Material property parameters used in the FEM simulation.

Material Parameters	Value
KOH (at temperature *T* = 26 °C)	
Dynamic viscosity (Pa s)	2.4
Density (kg/m^3^)	1305
Diffusion coefficient (m^2^/s)	3.79 × 10^−9^
Velocity of sound (m/s)	1988
Lead Zirconate Titanate (PZT-4)	
Density (kg/m^3^)	7500
Piezoelectric coefficient *e*_33_ (C/m^2^)	15.1
Relative dielectric constant *ε_r_*_33_	663.2
Structural steel	
Density (kg/m^3^)	7850
Young’s modulus (Pa)	200 × 10^9^
Poisson’s ratio	0.3
Nylon	
Density (kg/m^3^)	1150
Young’s modulus (Pa)	2 × 10^9^
Poisson’s ratio	0.4

## Data Availability

Not applicable.

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
