# Peer review of "Effect of Ultrasonic Excitation on Discharge Performance of a Button Zinc–Air Battery"

_micromachines, 2021, doi:10.3390/mi12070792_

Round 1
Reviewer 1 Report
The manuscript described the study on the effect of ultrasonic excitation on discharge performent of a button zinc-air battery. The data is interesting and is suitable to the journal. However, it need futher experiments or discussion so that the content of the research is clarified and the explanation of the obtained result is more proper.
1. Did the authors perform the ultrasonic excitation with only one set of condtion: 52.8 mm/s of vibration velocity and 161.2 kHz of working frequency. Can the authors change these parameters and show the influence of each parameter variation on the discharge performance?
2. How did the authors determine the materials parameters (viscosity, density, Poisson's ratio...) in Table 1? Or did the authors refer those data from some source. If the answer is yes, please provide the reference information.
And please correct following points:
- The style of writing a unit should be unified through the manuscript. For example, the authors wrote W.h kg-1 and mm/s. The former should be rewritten as W.h/kg. Please check all other units.
- Fig.2 (a) and Fig.5(a), "volatage" should be corrected to "Voltage"
- Fig.3 (b), "Chargr" should be corrected to "Charge"
- Fig.5 (c), "Open-circle" should be corrected to "Open-circuit"
Reviewer 2 Report
This is an interesting work. To be honest, I like the idea. Unfortunately, the subject is quite narrow and the way the subject is treated is not fully relevant, according to my knowledge.
The paper presents the findings related to the improved performance of a Zn-Air battery button type which capacity is improved by vibrating the battery at certain frequency. However, improved capacity of the battery is disputable as the authors did not detailed the procedure that indicated that improved performance. Modal analysis of any shape - including the ogive shape of the battery is quite simple as execution and to be honest, I missed why that modal analysis is relevant to the work. Details on this aspect are also missing in the paper. Another aspect: this paper reaches the energy aspects. Energy gain is a target for anyone. Here, if the battery is vibrated, a certain energy is required for that - which might be more that the gain through the increased capacity of the battery - such that overall power is lost. I would like to see this evaluation of the saved energy and the dissipated energy through vibration.
Some of the figures are not clearly explained.
Now, details:
Line 53: How the values were identified? Trial and error? Some modeling method? What kind?
Line 54 More details about this analysis are necessary to educate the reader
Line 71-2 How the capacity of the battery assessed?
Figure 2: On what kind of charge the assessment was carried out? Resistor? What resistance? I see, adjustable resistance box ZX21a
However, what was the resistance?
Line 100: what was the amplitude of the vibration?
Line 102: It may be possible that a lateral vibration may yield more efficient output? I would like to see some explanation of the phenomena that occur here - even if assumptions rather than describe the thing as a known fact
Line 108: is velocity or frequency of interest?
An explanation of the phenomenon is absolutely necessary
Line 111: Why this frequency? Velocity will be derived from the amplitude but why this frequency?
Line 152: I see limited need to investigate the vibration of the ring. The interest may be in the battery where this phenomenon of agitation occur
Table 1: Sure, there is a coupling but the output could be measured with the laser vibrometer. For the sake of modeling, the analysis is a good idea but mostly the interest is on what happens inside the battery - which is just skipped in this section
Line 167: This addition may be due to a piezoelectric phenomenon which amplified the electrochemical phenomenon.
Line 172-174 - the statement needs a reference or a solid proof, which the paper does not provide.
Conclusion: The paper is an excellent example of biased research. If we speak about batteries, we mean portable power. If power is not available, it must be carried, as in the case of batteries, If a battery is carried, we have limited power on us. If we need to power a piezoelectric ring, that will consume power from the available which will reduce the capability - time wise of the battery. I would have expected to have such a conclusion
Reviewer 3 Report
Manuscript address a Zn-air battery topic and the results are not absolutely new but interesting in my opinion. The conclusions are well supported and presented. However there are few comments and suggestions which I would like to address.
First comment is the fact that the authors are proposing a method for enhancing the battery efficiency and performances. They state that : “with the assistance of acousto-fluidic field, the discharge time is extended for about 3 hours, and the capacity is increased by 20%.”. However, the proposed solution is actually consuming electrical energy/power. Thus, for having a more relevant view for the reader I would propose the authors to also evaluate (in the discussions or conclusion paragraph) the energetic balance of the battery with the proposed acoustic device (e.g. ratio between the battery energy or power gain with the oscillator and the oscillator energy or power consumption).
Secondly is a comment of a result interpretation which does not sound to ‘intuitive’ to me:
“During the negative half periods of sound pressure, the distance among the molecules in the
electrolyte increases, causing that the cohesive forces among the molecules become small. During the positive half periods of sound pressure, the molecules are compressed by the positive sound pressure. However, the cohesive forces cannot increase too much because the molecules repel each other when they are too close. “
- since liquid are quasi-incompressible, the inter-molecular distance variation based on the pressure produced by the oscillation ring does not really sound to intuitive to me. Thus the statement should probably be better supported or rephrased.
And finally there are two more minor comments:
Introduction
“The experimental result shows that the output power of the battery increases from 22.2 mW to 32.6
mW, and the rating capacity increases from 330.56 mA·h to 396.89 mA·h when the vibration velocity is 52.8 mm/s at 161.2 kHz.”
- since the movement is an oscillatory one, the “vibration velocity” term should better be defined somehow to the reader.
- also, this phrase belongs to abstract and/or conclusions rather then introduction
“The analyses show that the ultrasonic effects in electrolyte solution, such as acoustic microstreaming vortices and viscosity decrease, contribute to the discharge performance improvement through enhancing the uniformity of OH - distribution and decreasing the resistance of mass transfer.”
- this phrase, unless is from bibliography and a reference should be indicated, is should rather be placed to the discussion part.
Results:
Fig. 2 - “The abbreviation v p-p is the averaged peak-peak value of the out-of-plane vibration velocity on the upper surface of piezoelectric ring. “ the explanation is welcome but should probably be presented inside the text and not as a figure caption.
Round 2
Reviewer 2 Report
Kindly please, answer to the concerns within the manuscript. I do not need explanations but the reader will. Hence, all the effort on the manuscript.

Round 3
Reviewer 2 Report
Congratulations for the newly accepted paper!